# A Current Landscape on Alport Syndrome Cases: Characterization, Therapy and Management Perspectives

**DOI:** 10.3390/biomedicines11102762

**Published:** 2023-10-12

**Authors:** Nahed N. Mahrous, Yahya F. Jamous, Ahmad M. Almatrafi, Deema I. Fallatah, Abdulrahman Theyab, Bayan H. Alanati, Suliman A. Alsagaby, Munifa K. Alenazi, Mohammed I. Khan, Yousef M. Hawsawi

**Affiliations:** 1Department of Biological Sciences, College of Science, University of Hafr Al-Batin, Hafr Al-Batin 39524, Saudi Arabia; n.mahrous@yahoo.com; 2The National Center of Vaccines and Bioprocessing, King Abdulaziz City for Science and Technology, Riyadh 12354, Saudi Arabia; yjamous@kacst.edu.sa; 3Department of Biological Sciences, College of Science, Taibah University, Madinah 42353, Saudi Arabia; a.m.almatrafi@gmail.com; 4Department of Medical Laboratory Sciences, College of Applied Medical Sciences, Prince Sattam Bin Abdulaziz University, Al-Kharj 11942, Saudi Arabia; n.fllatah@psau.edu.sa; 5Department of Laboratory and Blood Bank, Security Forces Hospital, Makkah 11481, Saudi Arabia; hawsa33@hotmail.com; 6Department of Biochemistry & Molecular Medicine, College of Medicine, Al-Faisal University, P.O. Box 50927, Riyadh 11533, Saudi Arabia; 7Center for Synthetic Microbiology, Bioinformatics Core Facility, University of Marburg, 35032 Marburg, Germany; bsyan.alanati@icloud.com; 8Department of Medicinal Laboratory Sciences, College of Applied Medical Sciences, Majmaah University, Al-Majmaah 11952, Saudi Arabia; s.alsaqaby@mu.eu.sa; 9Research Center, King Faisal Specialist Hospital and Research Center, P.O. Box 40047, Jeddah 21499, Saudi Arabia; munaifah29alenezi@gmail.com (M.K.A.); imrankhanitr@gmail.com (M.I.K.)

**Keywords:** Alport syndrome, type IV collagen, glomerular basement membrane, kidney disease, gene technology

## Abstract

Alport syndrome (AS) is a rare genetic disorder categorized by the progressive loss of kidney function, sensorineural hearing loss and eye abnormalities. It occurs due to mutations in three genes that encode for the alpha chains of type IV collagen. Globally, the disease is classified based on the pattern of inheritance into X-linked AS (XLAS), which is caused by pathogenic variants in COL4A5, representing 80% of AS. Autosomal recessive AS (ARAS), caused by mutations in either COL4A3 or COL4A4, represents 15% of AS. Autosomal dominant AS (ADAS) is rare and has been recorded in 5% of all cases due to mutations in COL4A3 or COL4A4. This review provides updated knowledge about AS including its clinical and genetic characteristics in addition to available therapies that only slow the progression of the disease. It also focuses on reported cases in Saudi Arabia and their prevalence. Moreover, we shed light on advances in genetic technologies like gene editing using CRISPR/Cas9 technology, the need for an early diagnosis of AS and managing the progression of the disease. Eventually, we provide a few recommendations for disease management, particularly in regions like Saudi Arabia where consanguineous marriages increase the risk.

## 1. Introduction

In 1875, Dickinson was the first scientist who reported the presence of familial inherited renal failure. In 1972, Arthur Cecil Alport was the first British physician who described symptoms of kidney and hearing health problems in families. He pointed out that females were less severely affected by deafness than males, although the disorder was more likely transmitted by females. Further cases were diagnosed, and SA was approved in 1961 as an eponym [1,2]. It has been observed that AS develops frequently in patients with hematuria (i.e., bloody urine), nerve deafness, edema, and hypertension [3].

Alport syndrome is significantly caused by mutations in one of three genes that code for type IV collagen alpha chains: COL4A3 (α3), COL4A4 (α4) and/or COL4A5 (α5) [4]. Those genes are involved in type IV collagen biosynthesis [5]. Type IV collagen mainly constitutes of the glomerular basement membrane (GBM), which represents ~50% of its total protein mass and is responsible for its stability. However, GBM is a thin extracellular matrix protein that functions as a per selective barrier to the passage of blood cells and proteins from the blood to the urinary tract [6] (Figure 1(3)). Changes in α-chains can give rise to dysfunctional GBM, leading to sensorineural deafness, abnormalities in several parts of the eyes, hematuria, proteinuria (i.e., excessive protein in urine) and eventual chronic kidney disease (CKD) [7,8].

Estimations of gene mutation frequency range from 1 in 5000 to 1 in 10,000, demonstrating its high burden in clinical settings [9]. As a rare disease, the estimated frequency of AS is approximately 1 in 50,000 live births worldwide [3]. Despite advances in therapeutic strategies that are performed to address many of the associated symptoms and slow the progression of kidney disease, there is no radical therapy for AS as of this moment.

Alport syndrome (AS) is a rare genetic disorder categorized by a progressive loss of kidney function, sensorineural hearing loss and eye abnormalities (Figure 1).

The aim of the current review is to provide updated knowledge about AS including its clinical and genetic characteristics in addition to available therapies that only slow the progression of the disease. Moreover, this review focuses on reported cases in Saudi Arabia and their prevalence. Additionally, we shed light on advances in genetic technologies like gene editing using CRISPR/Cas9 technology and the need for an early diagnosis of AS and to manage the progression of the disease. Eventually, we provide a few recommendations for disease management, particularly in regions like Saudi Arabia where consanguineous marriages increase the risk.

## 2. Clinical Characterization of Alport Syndrome

Clinically, AS characteristics are diverse, with a wide range progressing to end-stage renal disease (ESRD). The main and initial observed criterion among all the AS modes of inheritance is persistent hematuria, as approximately detected in all male and 97% of female patients [10]. Proteinuria is detected in male patients in early childhood, with levels gradually increasing with age, and occasionally presents as a nephrotic condition. It is also estimated that 90% of Proteinuric patients acquire ESRD by the age of 40 years, with the median age of ESRD development being 25 years. By the age of 40, 12% of females have acquired ESRD [10,11]. Also, Nozu et al. in 2019 reported that female XLAS patients developed proteinuria at a median age of 7 years and ESRD at a median age of 65 years [7]. Ultimately, 90% of male and 20% of female patients develop end-stage renal disease (ESRD) by the age of 40 years [11,12]. In the US, patients with various mutation types have significantly variable starting ages for ESRD. The following are the median (and 95% CIs) times from birth to end-stage renal disease: missense, 37 years (34–40); splice site, 28 years (26–32); truncating, 25 years (21–31); major deletion, 22 years (16–23); and small deletion, 22 years (19–27); P 0.0001 [13].

Further, ophthalmologic consequences include specific ocular abnormalities such as anterior lenticonus, posterior subcapsular cataract, posterior polymorphous dystrophy, and retinal flecks [13,14]. In addition, AS-affected patients frequently exhibit external disorders such as sensorineural deafness, of which frequently develops in late infancy; by the age of 40, 90% of male patients and around 12% of female patients have hearing loss [11,12] (Figure 2).

The majority of treatments for Alport syndrome involve a combination of angiotensin-converting enzyme inhibitors and angiotensin receptor blockers (ARBs) [14]. The research and development of medications for chronic renal illnesses, including Alport syndrome, have received increasing attention in recent years. Investigational medications such as sodium/glucose cotransporter 2 inhibitors (SGLT2is) [15], aminoglycoside analogs [16], endothelin type A antagonists [17], lipid-modifying medications, hydroxychloroquine (HCQ) [18], antimiR-21, bardoxolone methyl [19], and gene replacement therapy [20] are all in various stages of development.

## 3. Genetics of Alport Syndrome

There are six members of type IV collagen (COL4A1, COL4A2, COL4A3, COL4A4, COL4A5 and COL4A6), which encode six unique alpha chains of type IV collagen (α1–α6) (Figure 3).

Pathogenic variants in these genes can cause several progressive and non-progressive glomerular disorders [21]. Mutations in COL4A5 are the main cause of XLAS (Figure 2) and are responsible for more than 80% of AS cases [4]. It is located on chromosome Xq22 and has approximately 51 exons that encode 1685 amino acids [22]. The protein structure of the α5 chain includes three parts: A signal peptide with 26 amino acid residues, a collagenous domain with 1430 residues that consist of short non-collagenous interruptions and a 229 amino acid carboxy-terminal non-collagenous (NC) domain [11]. In 1990, Barker et al. reported the first X-linked inheritance of AS in three families, resulting in structural aberrations of the COL4A5 gene [23]. Since then, more than 1900 mutation sites in COL4A5 have been identified among AS patients, with approximately 1100 unique changes [18]. In X-linked disease, the most common mutations are missense mutations (40%) followed by splicing mutations (10%), nonsense mutations (7%), and frameshift and subsequent nonsense change downstream (30%) [24]. However, 12–15% of mutations in COL4A5 are “de novo” pathogenic variants [12,25]. The X-linked AS patients often exhibit a family history of hematuria, either with or without proteinuria, or might have renal failure. Hemizygous males with COL4A5 pathogenic variants exhibit more severe clinical manifestations than heterozygous females [21].

On the other hand, mutations in COL4A3 or COL4A4 genes are the main cause of autosomal recessive AS (ARAS) or autosomal dominant AS (ADAS) (Figure 2). Those genes are arranged head-to-head on chromosome 2 (2q36.3) and both share one promotor that encodes other type IV collagen genes [26]. The COL4A3 gene consists of 52 exons that span over 88 kb and encode 1670 amino acids, whereas COL4A4 spans 161 kb and is composed of 48 exons [4]. Both COL4A3 and COL4A4 proteins are composed of the N-terminal amino acid domain, the central triple-helical domain including G-X-Y repeats, and the C-terminal globular non-collagenous domain with a highly homologous protein structure with COL4A5 [27].

The ARAS mutations in either COL4A3 or COL4A4 are responsible for 15% of AS patients with no family history of AS [28]. Individuals with ARAS exhibit clinical features that are similar to those seen in males with XLAS. Carriers with the monoallelic variant are frequently asymptomatic or show only mild proteinuria and microscopic hematuria [24]. However, individuals with truncating variants are more likely to develop kidney failure before the age of 30 [28,29].

Additionally, there are no differences in incidence and clinical manifestations between male and females as this disorder sporadically appears in one generation [7].

In addition, pathogenic heterozygous variants in COL4A3 or COL4A4 genes can cause ADAS or thin basement membrane nephropathy (TBMN), which are often associated with hematuria and proteinuria [30] and might progress to cause late-onset kidney failure in nearly 25% of patients. It is a rare condition and has been recorded in 5% of all cases [23]. However, the severity of ADAS/TBMN can vary within patients, even among members from the same family [31]. It is reported that individuals with ADAS who carry heterogeneous truncating mutations progress to kidney failure at an earlier age compared to those carrying missense mutations [3].

Furthermore, digenic inheritance AS (DIAS) has been reported in some AS cases resulting from mutations in COL4A5 plus pathogenic variants in COL4A3 or COL4A4 [29,32] (Figure 2). Also, DIAS has been described as resulting from pathogenic variants in COL4A3 plus COL4A4 [33]. Male patients with DIAS in COL4A5 plus COL4A3 or COL4A4 might not show any worse evidence of the disease as all their collagen IV α3α4α5-heterotrimers affect the resulting pathogenic variant in COL4A5, whereas female patients with a pathogenic variant in COL4A5 plus COL4A3/COL4A4 are at a high risk, 75% of affecting their heterotrimers, thus increasing the risk of proteinuria [34].

The genotype–phenotype correlation has been studied in a European cohort [11]. Earlier genotype–phenotype correlation studies have revealed that males with large deletions, frameshift mutations, and truncating variants in the COL4A5 gene present with more severe clinical manifestations and are at a high risk of kidney failure at a younger age [11], whereas males with missense variants tend to exhibit milder disease features. In contrast, no clear genotype–phenotype correlation has been reported among females with heterozygous variants in COL4A5 and their age at kidney failure [12]. However, it has been observed that females with missense variants in the COL4A5 gene have better kidney function and are less likely to develop proteinuria in comparison to females carrying other types of variants [35]. A study by Bekheirnia et al. from 2010 examined the relationships between genotype and phenotype in a sizable US cohort of male patients with XLAS. As an outcome, they discovered that missense mutations (51% of the families) were most frequently found, followed by truncating (14%) and splice site (13.7%) alterations [13].

The most common pathogenic variants are the missense variants that affect Gly resides in the collagenous Gly Xaa Yaa repeats, which cause AS in Gly substitutions [4]. The clinical features and severity associated with Gly missense variants highly vary between patients [27]. In contrast, the mutation substitutions affecting non-collagenous boundary Gly residues resulted in kidney failure at a delayed age [4].

Recently, the 100,000 Genomes Project’s 39,421 participants were tested for the bioinformatically predicted harmful missense mutations and 5.6% of them had a history of hematuria [36]. In Saudi Arabia, due to the high consanguinity rate (58%), several recessive mutations were reported [37,38,39,40].

## 4. Cases of Alport Syndrome in Saudi Arabia

Kidney disease is the third most common disease in Saudi Arabia after obesity and breast cancer, with thousands of new cases identified each year. In April 2011, data from the World Health Organization (WHO) showed that 2540 deaths in Saudi Arabia (2.92% of all deaths) were attributable to kidney illness. Saudi Arabia is ranked 48th in the world with an age-adjusted death rate of 25.45 per 100,000 people [41].

Around 1997, AS was first discovered in Saudi Arabia. Some studies of Saudi children with kidney disease reported that AS is an infrequent diagnosis upon kidney biopsies and accounts for less than 4% of all cases. Molecular studies have highlighted common genes and the frequency of mutations that cause this syndrome in our population [42].

The prevalence of primary glomerulonephritis (GN) among people in Saudi Arabia’s western region was reported by Jalalah in 2009 [41]. The study was based on a retrospective analysis of renal biopsies that were archived during an 18-year period between 1989 and 2007. A total of 296 cases of primary GN were selected and analyzed by light, immunofluorescence and electron microscopy. The ages of the patients range from 17 to 76. These data showed that the membranous GN (MGN), which accounts for 25.7% of all primary GN cases, is the most common type, followed by focal segmental glomerulosclerosis (FSGSC), which accounts for 21.3% of cases. Immunoglobulin A nephropathy (IgAN) is a less frequent GN, accounting for 17.6%; membrano-proliferative GN (MPGN) represents 11.5%; immunoglobulin M nephropathy (IgMN) with 7.8%; minimum change disease (MCD) with 5.4%; and mesangio-proliferative GN (MesPGN) accounts for 4.7% of GN. Other rare types of GN included in this report are Fibrillar GN (FGN), accounting for 3%; post infectious GN (PIGN) with 2%; AS with 0.7%; and MPGN type II or dense deposit disease (DDD) with 0.3% [41]. The study showed that AS is a rare disease. Thus, more molecular and retrospective studies will be required to determine which area in Saudi Arabia is more abundant and to start treating and minimizing the risk of this syndrome.

## 5. Current Therapies for Alport Syndrome Management

Currently there is no Food and Drug Administration (FDA)-approved treatment for AS. All current therapies are non-specific and aim to slow down kidney decline and delay kidney failure. The current standard of clinical care can be divided into two categories: chemical drugs and molecular therapies. 

### 5.1. Chemical Drugs

According to the European Alport Registry, the renin angiotensin aldosterone system (RAAS) blocker is linked to a slower course of renal disease in heterozygous Alport carriers. Angiotensin-converting enzyme inhibitors (ACEIs) such as Ramipril can be used to treat heart failure and diabetic kidney diseases. An in vivo study showed that Ramipril with ARAS resulted in a significant reduction in various kidney diseases including proteinuria diseases [43]. Webb et al. in 2013 indicated that angiotensin receptor blockers (ARBs), such as losartan, were beneficial in controlling proteinuria [44]. The study also indicated that ACEI-type drugs show good results in children with AS.

Mineralocorticoid receptor antagonists (MRAs), such as finerenone, are widely used diuretics to reduce the severity of glomerulosclerosis, which is the hardening of the glomeruli in the kidney and renal interstitial fibrosis [45]. Research has also shown that finerenone is used to protect the kidneys and is effective in patients with type II diabetes and other kidney diseases such as polycystic kidney disease, renal artery stenosis and renal tubular acidosis. Aldosterone receptor antagonists, such as spironolactone, are also diuretics that can improve kidney function and reduce various chronic kidney diseases [46].

Sodium-Glucose Cotransporter-2 inhibitors (SGLT2i), such as Dapagliflozin, are FDA-approved drugs prescribed for patients who have type II diabetes. This treatment was considered to be immensely convenient in reducing the progression of chronic kidney failure and heart diseases [47]. A study of children with AS reported a 22% reduction in proteinuria after about three months of Dapagliflozin [48].

Metformin is a biguanide anti-hyperglycemic agent that can be used as first-line therapy to manage type II diabetes. It can reduce kidney fibrosis, inflammation and glomerular damage. The FDA has given Metformin a box warning and advises against using it in advanced renal disease (estimated glomerular filtration rate “eGFR” 30 mL/min/1.73 m^2^) due to its potential link to lactic acidosis [43].

Lipid-lowering agents such as statins are another treatment option. According to the Kidney Disease: Improving Global Outcomes (KDIGO) 2013 recommendations, people over 50 with an eGFR of less than 60 mL/min/1.73 m^2^ should take a statin. Patients with advanced renal disease are not covered by this (1A) [49]. It has been hypothesized that statin use may result in a slower course of renal disease [50].

### 5.2. Molecular Therapies

Short non-coding RNAs (<200 nucleotides) called microRNAs (miRNAs) can control the expression of genes by preventing or speeding up the degradation of the messenger RNAs they target [51]. Recently, several miRNAs have been introduced in clinical settings to be used for disease treatment, including miRNA-21. As well as other kidney diseases, the dysregulation of miRNA-21 has been discovered in AS. It was found that the use of anti-miRNA-21 oligonucleotides greatly slows the progression of kidney disease and increases survival in Alport mouse models after it was demonstrated that renal miRNA-21 is elevated in Col43/mice [52].

Several therapies improve outcomes in the animal models of AS including inhibitors of TGF-β1, vasopeptidase A and matrix metalloproteinases [53], BMP-7 [54], chemokine receptor 1 blockade [55], and stem cells [56]. However, none of these therapies have been prospectively researched in human AS populations.

Genome editing therapy is an experimental approach aimed at correcting faulty genes in order to treat a disease. It can be carried out using several methods, including turning harmful mutations dormant, inserting protective mutations, therapeutic transgenes or distorting viral DNA. The replacement of a mutant allele with a corrected copy of the gene necessitates the successful transport of the latter to an accessible tissue compartment through a carrier, such as a virus or nanoparticle [43].

## 6. Gene-Editing-Based Therapies

Gene therapy offers a bright prospect for the treatment of numerous illnesses and abnormalities [57]. An experimental treatment known as genome editing therapy tries to fix damaged genes in order to treat AS. Several techniques can be used to implement this new technology, including introducing protective mutations, changing viral DNA, rendering harmful mutations dormant, and using therapeutic transgenes [58].

With good in vitro outcomes in cases of homozygous thalassemia creating functioning red blood cell precursors, the Clustered Regularly Interspaced Short Palindromic Repeat (CRISPR/Cas9) system has emerged as a viable gene editing therapy for several uncommon genetic illnesses [58]. According to a recent study by Daga et al. in 2023, it becomes possible to acquire podocyte-lineage cells from urine that reproduce the physiological conditions encountered in these particular cells. This makes it prospective to accurately determine the COL4 variation correction index following experimental interventions [59]. Therefore, employing a self-inactivating dual-plasmid method produced by a self-cleaving streptococcus pyogenes Cas9 (SpCas9), the new technique was employed in AS autosomal dominant variants and X-linked hereditary AS. High correction rates were achieved, ranging from 44% in the COL4A3 gene to 58% in the COL4A5 gene, leading to lower indels (10.4% for COL4A3 and 8.8% for COL4A5) [52]. Despite these encouraging findings, because manipulating podocytes is challenging, there is still a long way to go before this proof of concept can be used within in vivo research.

Lin and colleagues in 2014 found that the podocyte release of 345 (IV) heterotrimers into a faulty GBM was successful at restoring the missing collagen IV network, which slowed the progression of renal illness and lengthened survival [60]. They employed a COL4A3/mouse model of AS and an inducible transgenic system to make this discovery. Through adenovirus-mediated gene transfer, Funk et al. in 2019, enhanced the expression of the COL4-3 transgene in the endothelium cells of COL4A3/Alport mice. The COL4A3/A4 or assembled 345 (IV) heterotrimer staining was absent in these AS mice, which prevented the resolution of the disease’s particular phenotype [61].

Despite the benefits of gene editing technology, there are several limitations and challenges. One of the challenges is the high cost of the technology. Also, the technology is not easily accessible and requires a high level of skill and training. Gene editing therapy has technical challenges with delivery, limited efficiency, off-target effects, and unexpected mutations. When CRISPR cuts the incorrect DNA sequence, it results in off-target effects that alter the genome in an undesired way. Unintentional mutations are haphazard mistakes that happen when the DNA is being repaired after CRISPR incisions. Due to CRISPR’s low efficiency, not all cells are properly altered, which lessens the intervention’s overall effectiveness. The delivery problems include difficulties in getting CRISPR components into target cells, particularly in complicated tissues and organs. The security, effectiveness, and caliber of CRISPR gene editing applications and products may be jeopardized by these technical restrictions.

## 7. Recommendations for Alport Syndrome Management

### 7.1. Kidney Transplantation

Mostly, patients with AS who develop ESRD need kidney transplantation. Therefore, it is advised to not consider kidney donations for family members who are heterozygous for COL4A3 and COL4A4 [34]. However, investigations revealed that with time, kidney function and proteinuria deteriorate in kidney donors with heterozygous pathogenic COL4A3 or COL4A4 mutations [62]. The development of anti-GBM antibodies without clinical manifestation is widespread, although post-transplant anti-GBM nephritis is an uncommon complication [63].

### 7.2. Genetic Counseling and Molecular Genetic Testing

Due to the high rates of consanguinity in the Saudi population [64], it is recommended that individuals with AS undergo genetic counseling to better understand the risks and inherited patterns associated with the condition. Consanguinity can increase the risk of developing ARAS in both sexes. It is recommended that mothers of an affected male with XLAS undergo urinalysis and genetic screening of the COL4A5 gene. Patients with TBMN should be screened for both COL4A3 and COL4A4 genes to exclude the pathogenic variant in COL4A5, particularly those with hematuria only [30]. Our previous article indicated that whole-exome sequencing was used to identify and characterize novel mutations in Saudi subjects’ chronic kidney disease (CKD) and autosomal dominant polycystic kidney disease (ADPKD) [40,65]. Digenic inheritance should be considered and might increase the risk of proteinuria, especially in affected females [29]. Families with an affected child are recommended to undergo genetic testing to identify pathogenic variants in the COL4A3, COL4A4 and COL4A5 genes before planning a new pregnancy. This would increase the chance of successful prenatal testing or preimplantation genetic diagnosis (PGD). Psychological supports should be provided to AS patients and their families to prevent the development of chronic mental health conditions [66,67].

Finally, genetic testing is a valuable tool for confirming the diagnosis of AS, especially in ARAS cases or when clinical features overlap with other kidney diseases. Whole exome sequencing (WES) is likely to be the most preferred diagnostic tool for AS since it examines all coding regions simultaneously [35].

## 8. Conclusions

For the past few decades, it has been assumed that AS is completely incurable. This syndrome is a major target for the drug industry to treat multiple diseases at the same time. However, there is a lot of research and experimentation going on to find treatments for AS that are needed to further improve their effectiveness. In Saudi Arabia, more genetic screening and testing is required. The specific priorities for future research based on the remaining knowledge gaps toward genetic studies are followed by gene therapies. More focus is needed on ARBs as a treatment option for Alport syndrome.

## Figures and Tables

**Figure 1 biomedicines-11-02762-f001:**
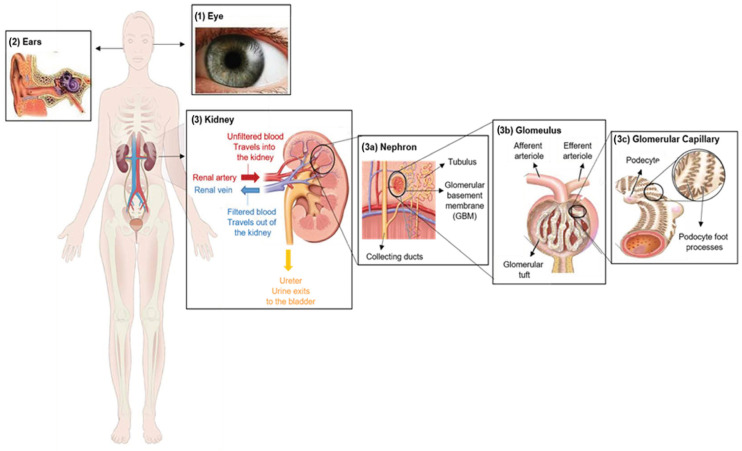
Clinical characteristics of Alport syndrome in individuals. Symptoms of Alport syndrome can be categorized by (1) eye abnormalities; (2) sensorineural hearing loss; and (3) related kidney symptoms.

**Figure 2 biomedicines-11-02762-f002:**
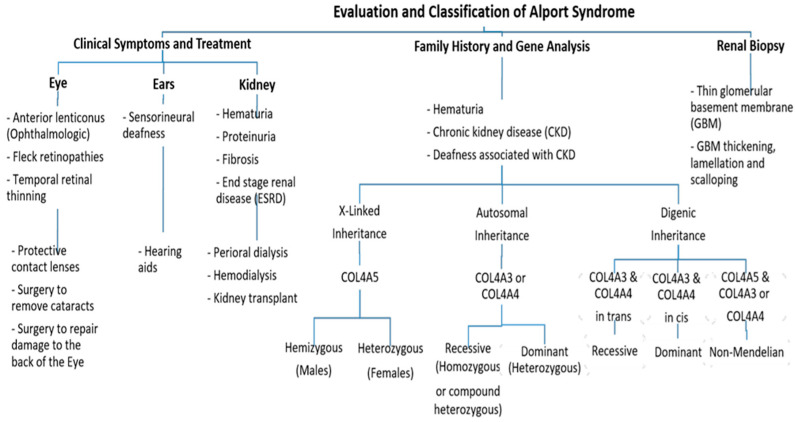
An approach regarding the evaluation and classification of Alport syndrome in individuals. In X-Linked Alport syndrome (X-Linked AS), the disease is usually passed from the mother to her child. In autosomal recessive Alport syndrome (ARAS), both sides of the family must have the mutation for AS to be passed on to their children, while in autosomal dominant Alport syndrome (ADAS), only one side of the family must have the mutation to pass it.

**Figure 3 biomedicines-11-02762-f003:**
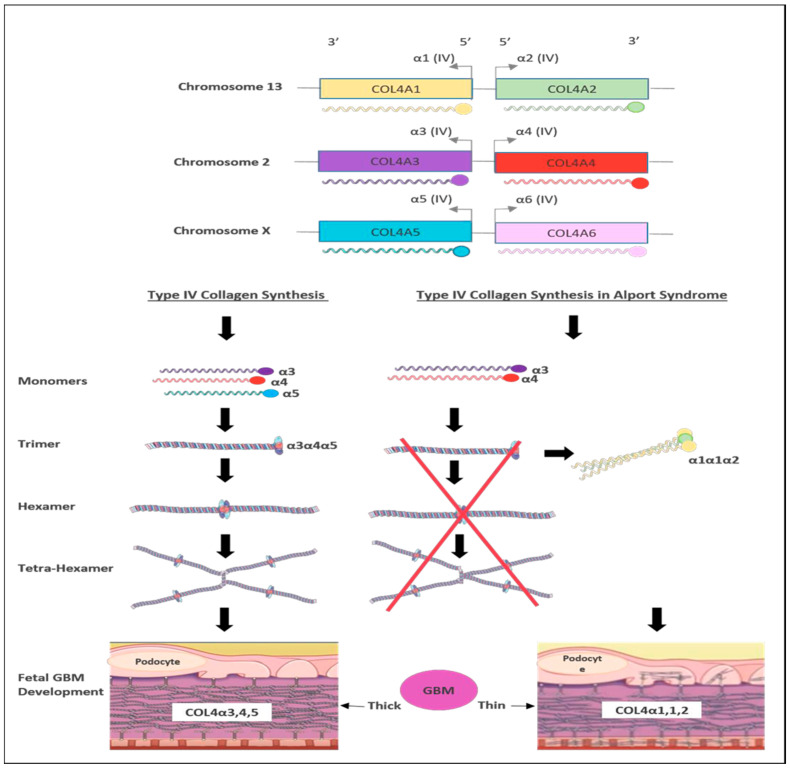
Schematic representation of type IV collagen biosynthesis. Type IV collagen genes are located in three different chromosomes pairwise, which encode six unique α-chains of type IV collagen (α1–α6). Monomers of collagen α3α4α5 can be combined among other forming triple helices (trimers). Two trimers then combine to form hexamers that can be associated with additional three hexamers that form tera-hexamer, which forms a thick GBM layer. However, in the absence of one monomer of the three α-chains, the kidney produces a monomer composed of α1α1α2 that eventually creates a thin GBM layer.

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
