# Peer review of "A Current Landscape on Alport Syndrome Cases: Characterization, Therapy and Management Perspectives"

_biomedicines, 2023, doi:10.3390/biomedicines11102762_

Round 1

Reviewer 1 Report

 Dear Editor thank you for the opportunity to reviwev  the manuscript titled "A Current Landscape on Alport Syndrome Cases: Characterization, Therapy, and Management Perspectives" provides a comprehensive overview of Alport Syndrome (AS), genetic disorder characterized by kidney dysfunction, sensorineural hearing loss, and eye abnormalities. The paper delves into the historical background of AS, its genetic basis, clinical manifestations, prevalence in Saudi Arabia, and current therapeutic strategies. It also discusses emerging gene-editing therapies and genetic counseling recommendations.

The manuscript presents valuable information on this complex condition. Here are some key points and feedback for the authors:

1. Comprehensive Coverage: The manuscript offers an extensive exploration of AS, from its historical origins to its current management strategies. This comprehensive approach provides readers with a thorough understanding of the condition.

2. Genetic Basis: The authors effectively explain the genetic basis of AS, including the mutations in COL4A3, COL4A4, and COL4A5 genes. This genetic insight is crucial for both clinicians and researchers working on AS. The addition of  table summarizing and depicting an overview of mutation types, clinical presentation, and phenotype could enhance the manuscript's visual appeal and help convey complex information more effectively.

3. Clinical Characterization: The manuscript provides a clear and detailed account of the clinical characteristics of AS, such as hematuria, proteinuria, and hearing loss. The inclusion of statistics regarding the age of onset and progression adds depth to the discussion.

4. Therapeutic Approaches: The manuscript outlines the current therapeutic approaches for AS, which include chemical drugs, molecular therapies, and gene-editing-based treatments. The discussion of emerging therapies, such as CRISPR/Cas9, is particularly noteworthy.

5. Genetic Counseling: The authors rightly emphasize the importance of genetic counseling for individuals and families affected by AS, especially in regions with a high prevalence of consanguinity like Saudi Arabia.

6. The manuscript concludes with valuable recommendations for AS management, such as kidney transplantation, genetic counseling, and molecular genetic testing. These recommendations are a useful guide for healthcare professionals.

7. Clarity and Structure: While the manuscript provides a wealth of information, it could benefit from improved organization and structure. Consider revisiting the flow of content to ensure a smoother transition between sections.

Overall, I believe this manuscript holds promise and provides valuable insights into the current state of knowledge about Alport Syndrome and it is my opinion that can be considered for publication.

Author Response

1         Reviewer -1

1.1        Open Review

Dear Reviewer-1

We would like to express our gratitude and appreciation for your fairly open review evaluation. There is no doubt in our minds that your valuable positive comments will definitely improve the overall quality of the article. We take your sincere advice in our concentration to improve our future manuscripts.

1.2        Comments and Suggestions for Authors

Dear Editor thank you for the opportunity to review the manuscript titled "A Current Landscape on Alport Syndrome Cases: Characterization, Therapy, and Management Perspectives" provides a comprehensive overview of Alport Syndrome (AS), a genetic disorder characterized by kidney dysfunction, sensorineural hearing loss, and eye abnormalities. The paper delves into the historical background of AS, its genetic basis, clinical manifestations, prevalence in Saudi Arabia, and current therapeutic strategies. It also discusses emerging gene-editing therapies and genetic counseling recommendations

Response:

Many thanks for your important encouraging feedback. We highly appreciate it. It is very kind of you.

The manuscript presents valuable information on this complex condition. Here are some key points and feedback for the authors:

  1. Comprehensive Coverage: The manuscript offers an extensive exploration of AS, from its historical origins to its current management strategies. This comprehensive approach provides readers with a thorough understanding of the condition.

Response:

Many thanks for this helpful comment.

  1. Genetic Basis: The authors effectively explain the genetic basis of AS, including the mutations in COL4A3, COL4A4, and COL4A5 genes. This genetic insight is crucial for both clinicians and researchers working on AS. The addition of table summarizing and depicting an overview of mutation types, clinical presentation, and phenotype could enhance the manuscript's visual appeal and help convey complex information more effectively.

Response:

Thanks for the valuable comment. The modifications are highlighted in yellow. We have included this information in Figure 2.

  1. Clinical Characterization: The manuscript provides a clear and detailed account of the clinical characteristics of AS, such as hematuria, proteinuria, and hearing loss. The inclusion of statistics regarding the age of onset and progression adds depth to the discussion.

Response:

Once again, many thanks for your recommendations and we agree with you. and is highlighted in yellow.

  1. Therapeutic Approaches: The manuscript outlines the current therapeutic approaches for AS, which include chemical drugs, molecular therapies, and gene-editing-based treatments. The discussion of emerging therapies, such as CRISPR/Cas9, is particularly noteworthy.

Response:

Many thanks for these important comments. CRISPR/Cas9 is advancing the chemical drugs, molecular therapies, and gene-editing-based treatments of alport syndrome.

  1. Genetic Counselling: The authors rightly emphasize the importance of genetic counseling for individuals and families affected by AS, especially in regions with a high prevalence of consanguinity like Saudi Arabia.

Response:

Many thanks for your comments. Agree with you that genetic counselling is important in disease management of Alport syndrome.

  1. The manuscript concludes with valuable recommendations for AS management, such as kidney transplantation, genetic counseling, and molecular genetic testing. These recommendations are a useful guide for healthcare professionals.

Response:

Many thanks for your constructive comments.

  1. Clarity and Structure: While the manuscript provides a wealth of information, it could benefit from improved organization and structure. Consider revisiting the flow of content to ensure a smoother transition between sections.

Response:

Many thanks for your helpful comments. We reviewed the manuscript and reorganised it.

Overall, I believe this manuscript holds promise and provides valuable insights into the current state of knowledge about Alport Syndrome and it is my opinion that can be considered for publication.

Response:

Thanks for your peer review and valuable comments on our research article entitled “A Current Landscape on Alport Syndrome Cases: Characterization, Therapy and Management Perspectives”. With great pleasure, we would like to inform you that the article has been revised comprehensively, and we addressed all your important comments. We hope our modification satisfies you and be considered for publication. Thanks for the positive comments and for your kindness.  

Reviewer 2 Report

This review article provides an overview of Alport syndrome (AS), a rare genetic disorder characterized by progressive kidney disease, hearing loss, and eye abnormalities. AS is caused by mutations in genes encoding type IV collagen, which is a major component of the glomerular basement membrane in the kidneys. The article discusses the clinical features, genetics, and inheritance patterns of AS. The most common form is X-linked AS, caused by mutations in the COL4A5 gene. Autosomal recessive and dominant forms are less common. The article also summarizes current treatments, which aim to slow kidney decline, and emerging genetic therapies like gene editing using CRISPR/Cas9 technology. Overall, AS remains incurable but better understanding of the genetics and clinical heterogeneity of the disease is leading to improved diagnosis and management. More research is still needed to develop targeted, disease-modifying treatments.

In summary, this review provides an overview of the clinical and genetic characteristics of Alport syndrome, a rare genetic disorder affecting the kidneys, hearing, and eyes. Current treatments only slow progression, so more research into genetic-based therapies like gene editing provides hope for more effective future management of the disease. The article also highlights the need for greater awareness and diagnosis of AS, particularly in regions like Saudi Arabia where consanguineous marriages increase risk.

  1. The article lacks a clear objective or thesis statement identifying the purpose and scope of the review. The introduction should state the focus and gaps this review aims to address.
  2. The clinical characterization section lacks detail on the range of phenotypes and severity of disease manifestations. More description of variability in progression and outcomes is needed.
  3. The genetics section is missing discussion of genotype-phenotype correlations between specific mutations and disease severity/progression. This could help guide prognosis and management.
  4. The treatment section mentions several pharmacological therapies but lacks details on dosing, efficacy data, and side effects. More thorough evaluation of treatment evidence is needed.
  5. Emerging genetic therapies are discussed but the article fails to critically assess challenges and limitations of technologies like gene editing in clinical applications.
  6. The Saudi Arabia cases lack population-level epidemiological data on prevalence and incidence of Alport in the region. The case reports summarized are anecdotal.
  7. Recommendations for management and screening are provided but not substantiated with evidence or expert guidelines. Supporting references are needed.
  8. The article concludes more research is needed but does not identify specific priorities for future research based on remaining knowledge gaps.
  9. As a review article, inclusion and critical appraisal of all relevant studies does not seem sufficiently comprehensive or systematic. A more rigorous methodology is needed.
  10. The article did not assess risk of bias, limitations, conflicts of interest or funding sources for cited studies. More disclosure and critical analysis of the evidence is required.

Author Response

1         Review 2

1.1        Open Review

Response:

We would like to express our gratitude and appreciation for your fairly open review evaluation. There is no doubt in our minds that your valuable comments will definitely improve the overall quality of the article. We take your sincere advice in our concentration to improve our future manuscripts.

1.2        Comments and Suggestions for Authors

This review article provides an overview of Alport syndrome (AS), a rare genetic disorder characterized by progressive kidney disease, hearing loss, and eye abnormalities. AS is caused by mutations in genes encoding type IV collagen, which is a major component of the glomerular basement membrane in the kidneys. The article discusses the clinical features, genetics, and inheritance patterns of AS. The most common form is X-linked AS, caused by mutations in the COL4A5 gene. Autosomal recessive and dominant forms are less common. The article also summarizes current treatments, which aim to slow kidney decline, and emerging genetic therapies like gene editing using CRISPR/Cas9 technology. Overall, AS remains incurable but better understanding of the genetics and clinical heterogeneity of the disease is leading to improved diagnosis and management. More research is still needed to develop targeted, disease-modifying treatments.

In summary, this review provides an overview of the clinical and genetic characteristics of Alport syndrome, a rare genetic disorder affecting the kidneys, hearing, and eyes. Current treatments only slow progression, so more research into genetic-based therapies like gene editing provides hope for more effective future management of the disease. The article also highlights the need for greater awareness and diagnosis of AS, particularly in regions like Saudi Arabia where consanguineous marriages increase risk.

Response:

Thanks for your peer review and valuable comments on our research article.  We highly appreciate your encouraging comments.

  1. The article lacks a clear objective or thesis statement identifying the purpose and scope of the review. The introduction should state the focus and gaps this review aims to address.

Response:

Again, thanks for this important comment. The aim and the scope of the review has been added in the introduction and highlighted in green.  

  1. The clinical characterization section lacks detail on the range of phenotypes and severity of disease manifestations. More description of variability in progression and outcomes is needed.

Response:

We highly appreciate your comments and recommendation. More description of variability in progression and outcomes has been added. The modifications are highlighted in green.

  1. The genetics section is missing a discussion of genotype-phenotype correlations between specific mutations and disease severity/progression. This could help guide prognosis and management.

Response:

Many thanks for your important comments. We agree with you and the manuscript has been updated accordingly. The authors discussed the genotype-phenotype correlations between specific mutations and disease severity/progression. Modifications are highlighted in green.

  1. The treatment section mentions several pharmacological therapies but lacks details on dosing, efficacy data, and side effects. More thorough evaluation of treatment evidence is needed.

Response:

Many thanks for this comment. The manuscript has been updated.

  1. Emerging genetic therapies are discussed but the article fails to critically assess challenges and limitations of technologies like gene editing in clinical applications.

Response:

Many thanks for the important comment. Sure, the technology of gene editing has some limitations and is indicated in the updated manuscript and highlighted in green.

  1. The Saudi Arabia cases lack population-level epidemiological data on the prevalence and incidence of Alport in the region. The case reports summarized are anecdotal.

Response:

Many thanks for your comments. The Saudi human genome program has not been completed yet. Therefore, precise data on the prevalence and incidence of Alport in the region is not available yet. Very limited research is ongoing.

  1. Recommendations for management and screening are provided but not substantiated with evidence or expert guidelines. Supporting references are needed.

Response:

Many thanks for your comments. More references were added.

  1. The article concludes more research is needed but does not identify specific priorities for future research based on remaining knowledge gaps.

Response:

Many thanks for your comments. specific priorities for future research based on remaining knowledge gaps has been added and highlighted in green in the conclusion section.

  1. As a review article, inclusion and critical appraisal of all relevant studies does not seem sufficiently comprehensive or systematic. A more rigorous methodology is needed.

Response:

Many thanks for your comments. The manuscript has been revised to ensure more rigorous methodology was used.

  1. The article did not assess risk of bias, limitations, conflicts of interest or funding sources for cited studies. More disclosure and critical analysis of the evidence is required

Response:

Once again, many thanks, in the updated manuscript the assess risk of bias, limitations, conflicts of interest or funding sources for cited studies was included.

Reviewer 3 Report

The article is interesting and comprehensively approaches Alport syndrome and current treatment options.
I actually have a few minor comments/questions for the authors.

In Figure 2, the abbreviations should be explained, preferably in the figure description.

I have a question whether the authors could present the motivation why they describe the cases of native patients in Saudi Arabia. Does this group have specific characteristics and differ from the typical patient in Europe? Are there differences in patient treatment? An interesting addition would be to add a patient management plan. Figure 2 shows only the classification.

There are no profound conclusions; those that exist are very general.
I think the abstract should not be as generalized as it is in the current version.

Sometimes the abbreviations appear twice, like "develop ESRD ESKD". This should be corrected.

My last question concerns the work methodology itself. Although this is not a systematic review, it would be helpful to have even very general information about which articles were taken into account, whether any were rejected, and what keywords the researchers used when searching the databases.

Author Response

1         Reviewer 3

1.1        Open Review

Response:

We would like to express our gratitude and appreciation for your fairly open review evaluation. There is no doubt in our minds that your valuable comments will definitely improve the overall quality of the article. We take your sincere advice in our concentration to improve our future manuscripts.

1.2        Comments and Suggestions for Authors

The article is interesting and comprehensively approaches Alport syndrome and current treatment options.

I actually have a few minor comments/questions for the authors.

Response:

Many thanks for your comments. We highly appreciate your positive comments and are willing to address your kind comments.

In Figure 2, the abbreviations should be explained, preferably in the figure description.  

Response:

Many thanks for your comments and we corrected Figure 2.

I have a question whether the authors could present the motivation why they describe the cases of native patients in Saudi Arabia. Does this group have specific characteristics and differ from the typical patient in Europe? Are there differences in patient treatment? An interesting addition would be to add a patient management plan. Figure 2 shows only the classification.

Response:

Many thanks for this valuable comment. One greater concern of AS cases in Saudi Arabia would be the consanguineous marriages. There is very limited information about AS abundant in Saudi Arabia as of the moment. Therefore, more studies are needed to determine the spread of the disease among the regions of Saudi Arabia and to start treating and minimizing the risk of this syndrome. Figure 2 shows the evolution and the classification of the disease through three means: symptoms, family history and analysis.

There are no profound conclusions; those that exist are very general. I think the abstract should not be as generalized as it is in the current version

Response:

Many thanks for your comments. We have added more information.

Sometimes the abbreviations appear twice, like "develop ESRD ESKD". This should be corrected.

Response:

Once again, thanks for your comments. We corrected the abbreviations.

My last question concerns the work methodology itself. Although this is not a systematic review, it would be helpful to have even very general information about which articles were taken into account, whether any were rejected, and what keywords the researchers used when searching the databases.

Response:

Many thanks for your comments. We have explained the cases of AS in Saudi Arabia in section 4 based on the available articles (references 47-50). However, data on the topic are old and limited as we mentioned, more studies are needed.